# Deformation Mechanism in Mechanically Coupled Polymer–Metal Hybrid Joints

**DOI:** 10.3390/ma13112512

**Published:** 2020-05-31

**Authors:** Karol Bula, Tomasz Sterzyński, Maria Piasecka, Leszek Różański

**Affiliations:** 1Institute of Material Technology, Faculty of Mechanical Engineering, Poznan University of Technology, PL-60965 Poznan, Poland; tomasz.sterzynski@put.poznan.pl (T.S.); maria.piasecka@student.put.poznan.pl (M.P.); 2Institute of Mechanical Technology, Faculty of Mechanical Engineering, Poznan University of Technology, PL-60965 Poznan, Poland; leszek.rozanski@put.poznan.pl

**Keywords:** polyamide 6, steel, hybrid joint, overmolding, deformation, IR thermography

## Abstract

In this, work, metal inserts were joined with polyamide 6 by using the injection-molding technique. The metal parts, made of steel grade DC 04, were mechanically interlocked with polyamide 6 (PA6) by rivets as a mechanical connection between both components in the form of s polymer filling the holes in the metallic parts. The mechanical-interlocking joints made of steel/PA6 were mechanically tested in a tensile-lap-shear test. The damage behavior of the joined materials in relation to rivet number and position on the metal plate was studied. The observation of rivet deformation was also conducted by infrared IR thermography. The study showed that, for polymer–metal joined samples with fewer than three rivets, the destruction of rivets by shearing meant sample damage. On the other hand, when the polymer–metal joint was made with three or four rivets, the disruption mechanism was mostly related to the polymer part breaking. The maximal values of the joint’s failure force under tensile-shear tests were achieved for samples where three rivets were used. Moreover, strong correlation was found between the surface temperature of the samples and their maximal force during the tensile-lap-shear test.

## 1. Introduction

Polymer–metal hybrid structures are promising energy-saving materials for manufacturing lightweight and high-mechanical-strength components, mainly for aerospace, automotive, railway, and household appliances. The joining processes of advanced hybrid materials usually involves mechanical coupling, while the molten polymer is formed by joining polymeric rivets in the through-holes of the metallic part [1,2,3,4]. With the production of metal–polymeric parts by means of injection molding, their features depend on those of the adhesion bonding. In this case, the creation of the surface topography of metallic parts using chemical etching or sandblasting with chemical functionalization is crucial for coupling quality [5,6,7,8,9].

Some interesting novel attempts for joining polymers with metallic parts were recently performed and published. The innovative solutions are mainly focused on microstructuring the metal surface before it comes in contact with the melted polymer, for example, by plasma etching, laser microabrading, and with the use of sophisticated joining processes like friction spot welding (FSW), friction riveting, joining by injection clinching, and ultrasonic joining—methods that are described below.

Drummer et al. [10] used plasma-etching pretreatment integrated into the injection-molding process by using a six-axis robot for handling the plasma nozzle. The novelty of this work is focused on an approach where both the thermoplastic part and the metal surface are simultaneously cleaned and etched by air plasma (in one mold); moreover, this should be done just before coinjecting the elastomeric interlayer. The integration of ionized-gas treatment with the application of an intermediate layer of a thermoplastic elastomer (TPE) led to significant reduction of process time.

Nanostructuring the surface of aluminum inserts was applied by Kimura et al. [11]. Since it was challenging to form the surface structures at a nanometer scale by using abrasive blasting or laser processing, chemical processing was applied via nanomolding technology [12] to create nanostructures on the surfaces of A5052 aluminum alloy pieces for joining experiments. As a result, a porous structure on the aluminum surface with pore diameters of 20 nm was achieved, and the structure consisted of a three-dimensional foam network like sponge foam. Together with surface modification, the influence of the molding parameters was observed, leading to the conclusion that the effect of injection speed was much higher than that of cavity pressure. The correlation between the level of surface roughness and bond strength, where the microstructure of the surface was manufactured by means of corundum blasting, was also experimentally tested by Lucetta et al. [13]. In contrast, in [14,15], describing the structures stamped in surfaces [15] by using corundum and acid [14], it was stated that an increase of surface roughness is not decisive on the creation of strength bonds.

Some papers were also devoted to presenting the laser-micropatterning technique, used as a novel tool to enhance micromechanical adhesion between metal and polymer, subjected to overmolding by injection [16,17]. Rodríguez-Vidal et al. [18,19] extensively studied the results of the application of various surface geometries produced by laser metal microstructuring and operating in two diverse modes of the laser beam. In this case, the impact of pulses with a length of nanoseconds was investigated, as well as the impact of a continuous wave on the failure force, at the conditions of the tensile-shear tests. Similarly, the influence of structural density and applied clamping pressure was also measured. It was found that, in the case of patterns shaped by laser operating in nanosecond pulse mode, the highest strength was achieved.

Another approach for microstructuring the hot-rolled AlMg3 alloy by electrochemical treatment, before joining with PA6 filled with 30 wt% of glass fiber, was presented in [20]. Surprisingly, the authors found that, to evaluate the strength of polymer–metal bonds at a direction perpendicular to the surface, roughness parameters were not chiefly applicable. They also verified that joint strength in such riblike specimens may by dependent simply on the existence of undercuts, which are usually necessary as an advantage of microstructuring treatment. The same assumption was presented in [21], demonstrating that each microstructure on the surface filled with molten polymer, oriented perpendicular to polymer flow and the tensile direction of the specimen, could form undercuts that are able to transmit the shear load.

Lambiase et al. [22] found that the mechanical properties of clinched joints between metal and polymer are practically equal to those between two metal parts. In the case where polymers characterized by high toughness are applied, for example, polycarbonate, preheating the arrangement is not necessary. So, the use of the technology of mechanical clinching is one of the fastest and economically advantageous joining procedures for assembling complex hybrid parts composed of metal and polymer components. The key features of polymers habitually applicable to serve as joints by clinching with the metallic part, by the use of a split tool set with a round profile, were described by the same author [23]. Therefore, studies of the impact on force joining and of thermal conditions on the opportunity to create connections with high mechanical performance by means of a simple clinching-tool arrangement, may be taken from these papers.

Lambiase and di Ilio [24] presented experimental investigations of the joining ability of aluminum with thermoplastic polymers by means of a split die with a round shape occupied by an exterior heater. The failure risk of such assemblies may be related to the use of a polymer characterized by a brittle fracture or by the ejection of the polymer from the joining.

Nowadays, the joining technique of a polymer to metals using rapid prototyping methods should also be considered. Ozlati et al. [25] ascribed fused deposition modeling (FDM), applied to the production of a lap connection of polypropylene (PP) fibers with Al–Mg alloy sheets placed between the PP fibers and as mechanical lock between the base aluminum sheet and the additive component. The impact of the area of the joint interface and of the primary heating of the components (by 20, 50, and 90 °C) on mechanical values was studied. Due to substrate preheating, improved strength of polymer-sheet bonds with the additive part was observed. Simultaneously, an improvement of printed-layer adhesion was noted.

Lastly, Feistauer et al. [26] provided an extensive review on polymer–metal direct-assembly methods. Generally, the benefits of assembled joints compared to bonds produced by adhesion and by mechanical clip were highlighted. A highly improved transfer of load from one material to another, lowering stress concentration, the skill to make much lighter connections, better performance on fatigue charges, improved damage tolerance and surface finish, as well as the advantageous distribution of stress are among the most chief arguments for such joints. On the other hand, the authors listed some crucial restrictions, including differences in the thermal expansion of the materials, leading to the creation of residual stresses and the result that joints cannot be disassembled. Nondestructive-technique (NDT) inspection is appropriate in this case, but the assembling cycle is frequently time and labor consuming.

The purpose of our studies was to record and analyze the deformation behavior of the rivetlike mechanical coupling of a metal plate with PA6, in relation to rivet number and position on the metal plate. Rivet position was measured from the edge of the metal insert that was first filled by a molten polymer. The IR thermography technique was applied as a noncontact and nondestructive tool for the determination of the heat transfer in the area of rivet connections/plastic joints. The observation of rivet deformation by means of an IR camera provides additional information concerning the initiation and development of plastic deformation, followed by crack propagation. The mechanical data were considered in relation to the corresponding IR thermograms.

## 2. Materials and Methods

### 2.1. Metal-Insert Preparation

The PA6 injection molding grade Tarnamid T-27, with a melt volume flow index (MFI) of 120 cm^3^/10 min at 275 °C (supplied by Grupa Azoty, Tarnów, Poland) and its composite with talc, a mineral filler with a shallow form, suitable to use as markers of the liquid-flow lines, were applied in our investigations. The creation of PA6/talc composites was described in the previous paper [27], dealing with analysis of polymer flow at the injection mold with a metal insert, where talc particles were used as markers of molten-polymer streaming vectors. The metal insert made of low-carbon steel no. 1.0338 (grade DC 04) EN 10130 was used as a partner in the adhesion connections. The dimensions of the metal insert was 75 × 10 mm (length × width, respectively), and the thickness was 1 mm.

Holes with a diameter of 3 mm, pierced in the metal parts, with numbers 1, 2, 3, or 4, were placed at a position of 10 mm from the edge of the plate. The distance of 10 mm was kept between the centers of the holes (see Figure 1a). Figure 1b shows a picture of a single drilled hole.

Hydrochloric acid (HCl) with a concentration of 5% was used to create the microstructuring surface of the metal plate. Prior to etching with acid, the surfaces of the metal plates were carefully cleaned with acetone. The roughness parameters of the metal insert surface after etching was Ra = 1.7 µm and Rz = 13.37 µm. Before injection molding, the surface of the metal sheets was cleaned again with acetone, using a cleaning cloth to remove the rest of possible surface contaminations.

### 2.2. Injection-Overmolding Procedure

The polymer–metal hybrid samples were produced by means of the ENGEL 20/80 HLS injection-molding machine (Engel Austria GmbH, Schwertberg, Austria), with a screw diameter of 22 mm. A multicavity mold with cavities of the same flow length (150 mm total flow length, 10 mm width), and thickness of 1 mm was utilized. The considered metal plates with holes were placed at the cavity opposite the gate position, as is shown in Figure 2, and kept in the closed mold for 30 s to raise the temperature to 60 °C. Subsequently, filling the cavity with PA6 composites was realized at 80 °C mold temperature, with an average injection rate of 90 mm/s. Temperature along the barrel was 225, 230, 240, and 260 °C, and injection pressure was set at 7 MPa. By filling the cavity, the front of the molten polymer met the edge of the metal insert at half the whole flow distance; thus, polymer pressure at the insert contact area was slightly reduced compared to injection pressure at the beginning of the filling process. After cooling termination, the test specimens were manually removed from the mold, and mold-ejector movement was disabled to avoid sample destruction.

### 2.3. Mechanical-Test Conditions, Thermography, and Microscopy Observation

Tensile tests were completed according to the EN ISO 1465 standard [28] by means of a ZwickRoel Z010 (Ulm, Germany) universal testing machine, operating under All Suite software, with a 10 kN load cell. Samples with single lap-joint geometry were designated to determine the strength of joints by shear test at a crosshead speed of 2 mm/min. To record the way of polymeric-rivet deformation (positions where the polymer was mechanically coupled to the metal plate) during uniaxial elongation, infrared camera FLIR X650sc (FLIR Systems, Inc., Wilsonville, Oregon USA); resolution 640 × 512 pixels, temperature resolution 0.02 °C) was used (see Figure 3a). Our task was to record the evolution of temperature rise around the rivet joints and to compare the surface temperature with the mechanical response of single-pin deformation under shear forces. To increase the surface-emissivity factor by IR observation, the polymeric surface of the samples was painted with a black nonreflective colorant (see Figure 3b).

To analyze the molten-polymer flow behavior around the rivets during mold filling, talc particles were used as flow markers. The flow nature of polymer macromolecules was detected by the statement of talc-particle allocation using polarized optical microscope (POM) Nikon Eclipse E400 (Nikon Corporation, Tokyo, Japan). Samples of PA6/T10 for this study were cut using rotary microtome Leica 256 RM (Leica Biosystems Nussloch GmbH, Nussloch, Germany) with a thickness of 20 μm along the flow direction, perpendicular to the specimen plane.

## 3. Results and Discussion

### 3.1. Lap-Shear Test Results

Mechanical testing of polymer–metal joints with rivets was performed at the regime of the lap-joint-shear test. Because of the divergence of the sample form, comparing with lap-shear-test standardization, the maximal force reached by the specimen break was taken as the equivalent value needed to assess the influence of rivet numbers on metal–polymer joint properties. In Table 1, the maximal force measured by the tensile-shear test is presented. It was found that the deformation mechanism of polymer–metal joined samples depended on the number of rivets. Thus, the maximal force values were registered in the case of the separation of joints with at least three rivets. The destruction of polymer–metal joint samples caused by polymer fracture was only observed in the case when the polymer–metal joint was made with three or four rivets. In samples with fewer than three rivets, deformation was noted in the rivet pull-out way or incidentally by the shearing of the rivets. Therefore, the value of breaking force listed in Table 1 is not always related to the fracture of the polymeric part.

Ozlati et al. [25] studied bimaterial joints realized with only one rivet formed by FDM, with various diameters. On the basis of the tensile-shear test, it was found that all samples failed at the joint area, which meant that only shearing of the rivets had occurred.

Clear local deformation was observed in samples joined with three or four rivets, followed by a break localized under the first rivet. The fracture of samples with three or four rivets was always localized at the same area and was preceded by the necking of the polymeric part, signifying that, in the case of the connection with three rivets, a force level leading to the plastic deformation of polyamide was needed. Thus, during the formation of necklike deformation, an entanglement of a polymer macromolecule structure may take place, so the material became tough in this region, followed by strengthening the elongated polyamide. This effect may have led to the fracture at the bottom edge of the rivet, whereas the cross-section continuously decreased in size due to the significant plastic deformation (see Figure 4a). Thus, a possible explanation of the localization of polymer breakage may be related to the existence of evident cross-section changes around the rivet (see Figure 4b,c).

On the other hand, more probable is the local structure weakness of the rivet resulting from molten-polymer flow instabilities by mold filling [27]. This effect may be related to the disarrangement of fountain flow in the proximity of rivets [29,30], in other words, the local disorientation of the macromolecular structure by molten polymer filling the cylindrical holes may occur. This occurrence was found by observing polymer-flow instabilities in the flow channel with a reduced cross-section, was also revealed and proved by observations done using flow markers like anisotropic inorganic fillers (mica, talc, short fibers) [31,32,33].

Flow-line distribution in the rivet region was attained by optical visualization of the polymer flow by mold filling. In this case, markers in the form of talc-like plates with a concentration of 10 wt% were introduced into the PA6, and the microtome samples of the polymer taken at the rivet section were observed by means of polarized optical microscopy. Dark short lines corresponding to the oriented talc particles may be seen in Figure 4b,c. The presence of flow markers frozen in the polymeric matrix in the form of talc particles, placed perpendicularly and almost parallel to the flow direction, allowed the conclusion that, in this area, fairly turbulent flow rather than fountain-like behavior prevailed.

### 3.2. IR Thermography Validation of Tensile Tests

The IR camera technique was applied to follow the deformation mechanism of the tested specimens at the area of the joint lap during the shear test. The IR scan of samples with a corresponding temperature profile, generated due to the self-heating effect during the elongation test, is presented in Figure 5. Moreover, temperature traces recorded at the area where the highest deformation of the polymer matrix took place are presented next to the thermographs. These regions correspond to locations centered on the most significant rivet-neighboring deformation area.

In Figure 5a, there is a thermography picture of an elongated specimen recorded just before specimen damage. Moreover, the plot of temperature development in the rivet area clearly indicates the most deformed area of the specimen, as well as the moment when the polymer was detached from metal, corresponding to the rivet pull-out. Due to small local polyamide deformation, there was no polymer breaking or rivet shearing; thus, the recorded temperature rise was relatively small.

Quite similar temperature runs may be seen for the sample coupled with two rivets (Figure 5b), where temperature peaks indicate the damage of the specimen, localized at the rivet positions. The measured temperatures at both rivet localizations were about 3.5 °C higher compared with that of the specimen surface. This rather low temperature shift appearing during specimen break depended on the type of deformation mechanism of samples joined with two rivets, in other words, when a rivet-shear process takes place.

Fully unlike the above situation was the breakage of samples with three or four rivets. The highest temperature value, recorded by the IR camera during the tensile-lap-shear test, was noticed at the area below the first rivet, which is shown in Figure 5c,d. Moreover, its values to some extent coincided with the uppermost forces measured during the tensile test. This means that heat radiation revealed by IR device, localized in the area of sample breakage, was strongly related to the nature of sample deformation. In the case of the sample coupled with three or four rivets, we saw that specimen deformation was initialized under the first rivet and further developed, and a process of necking formation occurred.

To highlighting the ability of the IR technique to register local plastic deformation, IR thermograms recorded for samples with four rivets are presented in Figure 6a–d, shown as an example of four consecutive stages, where evolution of the neck formation by elongation took place. For specimens where necking occurred, temperature traces were recorded for a longer period (over 10 min; see Figure 5d) than for specimens without extensive plastic deformation. Remarkably, the process of neck formation did not run uniformly. As was expected, deformation development started under the first rivet (see Figure 6a). Next, the warmest area of the deformed material shifted to the bottom part of the PA6 specimen, outside the coupled zone (see Figure 6b). The third analyzed deformation stage corresponds to the simultaneous development of structure fibrillation in PA6. In this stage, the necking effect was localized in both parts of the specimen (see Figure 6c), brought in some way to the local strengthening of the specimen, and followed up by crack initiation localized under the first rivet. Finally, crack propagation, which is well seen as a double hot point on the IR thermogram (Figure 6d), took place, and the polymer part was damaged just under the first rivet, which is illustrated in Figure 6e. This phenomenon was easily detectable by IR thermography, especially the area of deformation initializing. Moreover, such unusual relocation of the necking process was first observed in real-time measurements in this work. Additionally, the change in heat signal, detected from the surface of the specimen that was under tension, from a single one to separate signals, may prove the existence of a crack or void in the deformed element, so it may indicate an approach of damage occurrence, as was proven in our research. Therefore, it may be stated that an IR camera can be used as a tool in the monitoring of light hybrid constructions that are under tension. As was proven, thermography may be a crucial method in the detection of the weakest points in such bimaterial structures or elements.

## 4. Conclusions

In this work, we mechanically tested coupled pieces/elements made of two different materials, PA6 and steel. During tensile tests, the deformation behavior of the rivet-like mechanical coupled PA6/steel in relation to rivet number and position on the metal plate was investigated.

The most pronounced observation coming from our investigations is that the nature of fracture mechanics of injection molded plastic–metal parts joined with rivets is considerably dependent on the number of rivets. It was found that the mechanical joints of PA6 with a metal insert coupled with three rivets are the most valuable.

Correlation between sample-surface temperature and maximal sample-breaking force was found. Additionally, as was shown, IR observation of polymeric specimens under tension enables the recognition of the thermal signal that is responsible for crack propagation. Thus, the IR thermovision technique is helpful for the detection and monitoring of the deformation mechanism of sample areas where a potential fracture of the polymer structure may begin, in other words, it may serve as a supporting tool to reveal the premature effects of polymer–metal damage processes.

## Figures and Tables

**Figure 1 materials-13-02512-f001:**
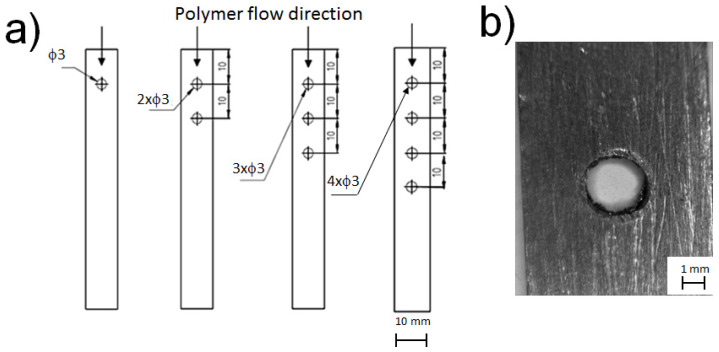
View of polymer-metal connection. (**a**) Hole allocation on metal plates, (**b**) example of a single hole.

**Figure 2 materials-13-02512-f002:**
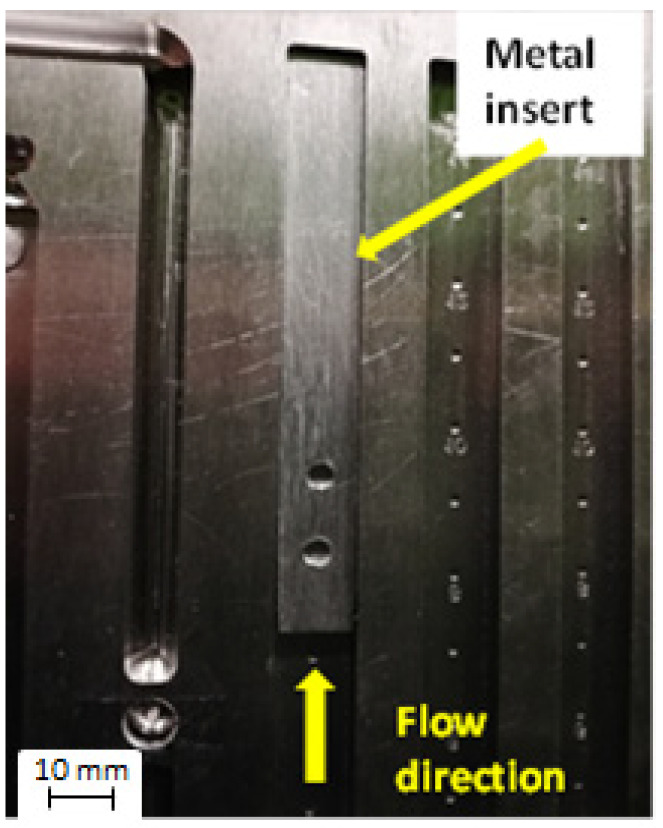
Injection-mold cavity with attached metal insert.

**Figure 3 materials-13-02512-f003:**
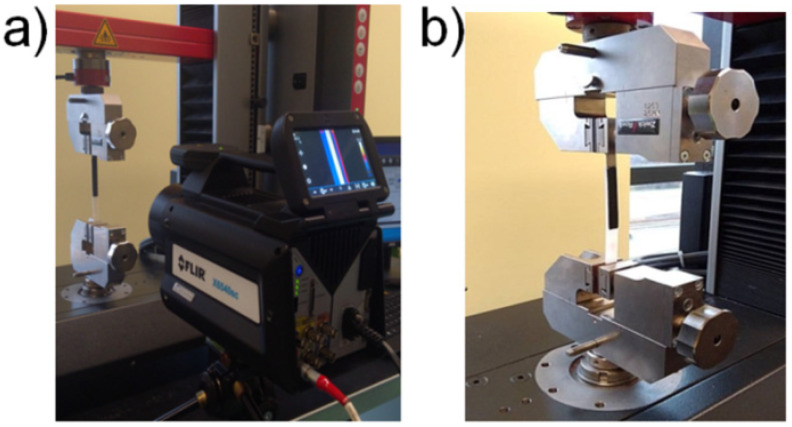
IR camera position versus sample face mounted in (**a**) clamping units; (**b**) mounted sample, black surface indicates polymer part in lap-joint sample.

**Figure 4 materials-13-02512-f004:**
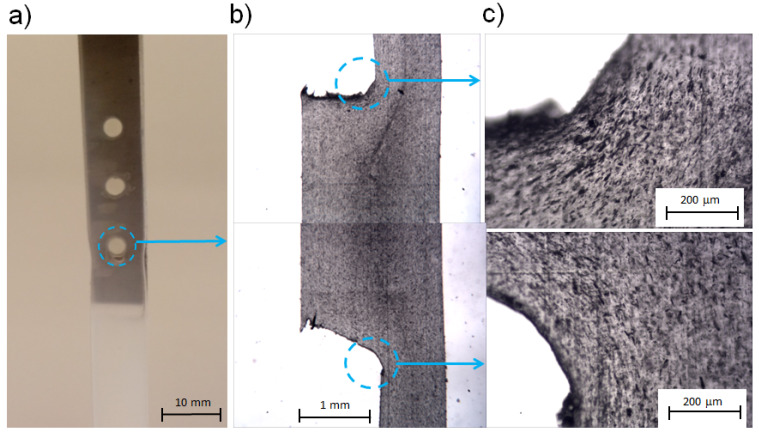
(**a**) Specimen with visible neck formed during uniaxial deformation, neck formation and crack localization under 1st rivet; (**b**) cross-section of single rivet where talc filler was used as polymer-flow marker; (**c**) visualization of talc-particle arrangement (dark short lines) in sample region located under first rivet.

**Figure 5 materials-13-02512-f005:**
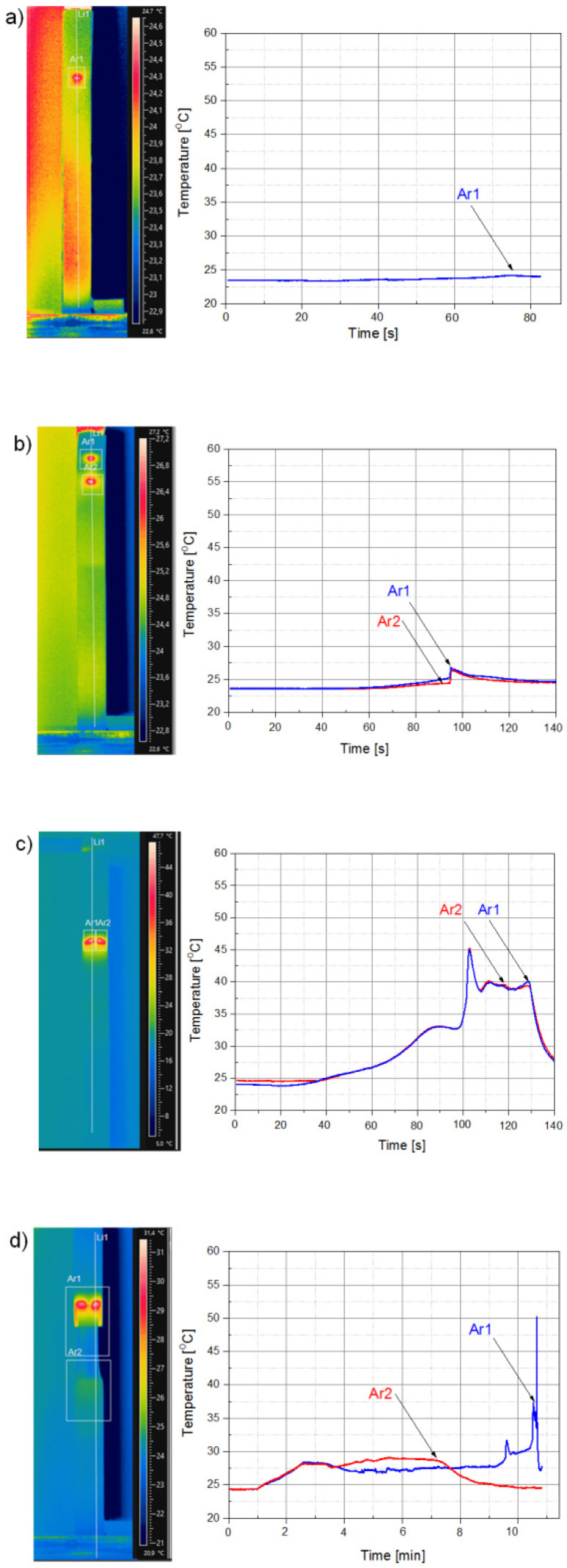
(**a**–**d**) IR thermograms for samples charged with maximal force during lap-shear test, and temperature traces adequate to rivet position in each sample from first to fourth rivet, respectively. Traces Ar1 and Ar2 illustrate evolution of maximal temperature recorded in sample area, indicated as Ar1 or Ar2, marked on each thermogram.

**Figure 6 materials-13-02512-f006:**
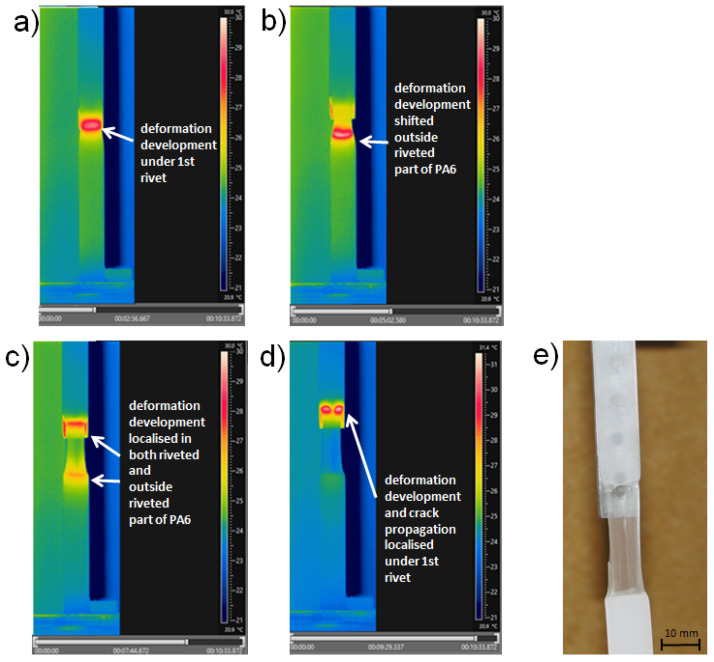
IR thermograms for sample coupled with four rivets during lap-shear test, pictures viewed in time sequence after test began: (**a**) 2 min 56 s; (**b**) 5 min 2 s; (**c**) 7 min 44 s; and (**d**) 9 min 29 s. (**e**) Picture of sample with neck and crack formation.

**Table 1 materials-13-02512-t001:** Mechanical results of single lap-joint test for samples with rivets with information about nature of sample damage. For each tensile test, six samples were used.

Sample PA6/DC04 Single-Lap Joint	Maximal Force (N)	Standard Deviation (N)	Tensile strength (MPa)	Standard Deviation (MPa)	Maximal Temperature ^2^ (°C)	Specimen-Deformation Behavior
One rivet	199 ^1^	35	−	−	25.4	Rivet pull-out
Two rivets	476	93	−	−	27.5	Rivet shear
Three rivets	618	55	65.1	5.7	48.0	Polymer part break
Four rivets	606	14	63.7	1.5	55.0	Polymer part break

^1^ Most tested specimens were damaged via detachment between polymer and metal pieces. ^2^ Temperature recorded during lap-shear tests by thermography camera.

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
