# Peer review of "Deformation Mechanism in Mechanically Coupled Polymer–Metal Hybrid Joints"

_materials, 2020, doi:10.3390/ma13112512_

Round 1

Reviewer 1 Report

The topic presented is important for the production of safe and reliable hybrid joints in lightweight constructions. Although the results are well described, there are some comments on the manuscript, these are:

Abstract: metal inserts was joined -  please check the grammar.

The abbreviation PA6 is not explained after the first mention, for example onthis way: Polyamide 6 (PA6)

Line 161 Microscpy Observation

Line 167 piks.,temperature

The standard deviation of force (N) is given in Table 1. Please indicate how many samples were tested for each parameter set.

Also there is no information on how the tensile force (N) achieved correlates with the real strength properties of the materials used. Please provide further data on what to expect as a strength value for the hybrid joint tested.

Scale bar in the images of samples in Fig 1b, Fig 2, Fig 4 is missing.

The graphic results in Fig. 5 are difficult to interpret due to the low image resolution. A revision of images to improve the image quality / resolution is recommended. Scale is missing in Fig 5a and Fig 5b.

Fig 6. Please indicate in the legend or text what Ar1 and Ar2 are.

The tensile test in Fig 6d takes almost 11 minutes compared to the 1 or 2 minutes in the other tests. Please comment this effect.

The temperature curve in Fig 6a is quite noisy. Is it possible to display signals in the same signal quality?

The adjustment of the temperature scale to a maximum of 60 °C for Fig 6a, b, c, d would enable better comparison of the measured temperature peaks. Please adjust it if possible.

Another recommendation is to check in the author guidelines whether the use of passive or personal pronouns (we, ours) is preferred. To my knowledge, nowadays passive is commonly used in academic writing. But there are different rules. Please check it.

Author Response

Response to Reviewer 1 Comments

The topic presented is important for the production of safe and reliable hybrid joints in lightweight constructions. Although the results are well described, there are some comments on the manuscript, these are:

Point 1: Abstract: metal inserts was joined -  please check the grammar.

Response 1: The changes have been made in the manuscript and is marked in yellow. Line no. 14,15

In this, work, metal inserts were joined with polyamide 6 by using the injection-molding technique.

Point 2: The abbreviation PA6 is not explained after the first mention, for example on this way: Polyamide 6 (PA6)

Response 2: The change has been made in the manuscript and is marked in yellow. Lines no. 15-17

The metal parts, made of steel grade DC 04, were mechanically interlocked with polyamide 6 (PA6) by rivets as mechanical connection between both partners in the form of polymer filling the holes in the metallic parts.

Point 3: Line 161 Microscpy Observation

Response 3: The change has been made in the manuscript and is marked in yellow. Line no. 163

Mechanical-Tests Conditions, Thermography and Microscopy Observation

Point 4: Line 167 piks.,temperature

Response 4: The change has been made in the manuscript and is marked in yellow. Line no. 169

camera FLIR X650sc (USA; resolution 640 x 512 pixels, temperature resolution 0.02 OC) was used.

Point 5: The standard deviation of force (N) is given in Table 1. Please indicate how many samples were tested for each parameter set.

Response 5: The appropriate information has been added in the manuscript and is marked in yellow. Line no. 204

 For each tensile test, six samples were used.

Point 6: Also there is no information on how the tensile force (N) achieved correlates with the real strength properties of the materials used. Please provide further data on what to expect as a strength value for the hybrid joint tested.

Response 6: The appropriate information’s have been added into the Table 1, and are marked in yellow.

Point 7: Scale bar in the images of samples in Fig 1b, Fig 2, Fig 4 is missing.

Response 7: The images have been supplemented with scale bars, and they are pasted in the manuscript.

Point 8: The graphic results in Fig. 5 are difficult to interpret due to the low image resolution. A revision of images to improve the image quality / resolution is recommended. Scale is missing in Fig 5a and Fig 5b.

Response 8: The image was modified and replaced as a Figure 4.

In revised version Figure 5 is presented as Figure 4.

Point 9: Fig 6. Please indicate in the legend or text what Ar1 and Ar2 are.

Response 9: The additional information has been added in the legend on Figure 5  and is marked in yellow. Lines no. 277-279

In revised version Figure 6 is presented as Figure 5.

Traces Ar1 and Ar2 illustrate evolution of maximal temperature recorded in sample area, indicated as Ar1 or Ar2, marked on each thermograms.

Point 10: The tensile test in Fig 6d takes almost 11 minutes compared to the 1 or 2 minutes in the other tests. Please comment this effect.

Response 10: The additional explanation has been added in the manuscript and is marked in yellow. Lines no. 284-285

In revised version Figure 6 is presented as Figure 5.

For specimens where necking occurred, temperature traces were recorded for a longer period (over 10 min; see Figure 5d), instead of specimens without extensive plastic deformation.

Point 11: The temperature curve in Fig 6a is quite noisy. Is it possible to display signals in the same signal quality?

Response 11: The correction in the figure have been done and introduced into the manuscript In revised version Figure 6 is presented as Figure 5.

Point 12: The adjustment of the temperature scale to a maximum of 60 °C for Fig 6a, b, c, d would enable better comparison of the measured temperature peaks. Please adjust it if possible.

Response 12: The corrections in the figure have been done and introduced into the manuscript

In revised version Figure 6 is presented as Figure 5.

Point 13: Another recommendation is to check in the author guidelines whether the use of passive or personal pronouns (we, ours) is preferred. To my knowledge, nowadays passive is commonly used in academic writing. But there are different rules. Please check it.

Response 13: Grammar check of the whole manuscript has been performed by MDPI English Editing service.   

Reviewer 2 Report

This article discusses the properties of the connection between metal inserts and polyamide by using injection molding technique. In general, the article was done quite accurately and technically competently; the article is of certain scientific interest. We can recommend that the authors pay more attention to metallographic studies of the nature of the compound (including the study of thin sections using SEM), the study of the nature of the destruction of internal stresses, etc. Such additional research could enhance the scientific relevance of the study.

Some recommendations to the authors:

The abstract as a whole is clearly formulated and gives an idea of the content of the article.

The second paragraph of the introduction - lines 40-41 - it is desirable to add appropriate references, for example, "for example by plasma etching [***], laser micro abrading [***], ..."

Line 44 and then - I recommend that you specify a reference immediately after the author of "Drummer et al. [***] used the ..."…” Rodríguez-Vidal et al. [***]…” etc.

Line 115 - not sure if quotes are needed in the word “filled”

Figure 1.а - technical image quality should be improved.

Figure 3. - at least image a is not in my opinion redundant. In principle, the general view of the equipment does not give much to the reader. A circuit diagram would be preferable.

Table 1. - it is desirable to use a decimal point, not a comma to indicate the decimal point.

Figure 4. - I recommend increasing the magnification - now these images do not provide much information.

A large number of quoted terms is not very good for a scientific style ... “rivet local”, so-called “classic” fountain (it is not clear why “so-called” - who calls it that), so-called “touch-off” , “Frozen”, in such “bi material coupling” - in most cases, quotation marks are simply superfluous.

Figure 5 - image quality should be improved

Author Response

Response to Reviewer 2 Comments

This article discusses the properties of the connection between metal inserts and polyamide by using injection molding technique. In general, the article was done quite accurately and technically competently; the article is of certain scientific interest. We can recommend that the authors pay more attention to metallographic studies of the nature of the compound (including the study of thin sections using SEM), the study of the nature of the destruction of internal stresses, etc. Such additional research could enhance the scientific relevance of the study.

Some recommendations to the authors:

Point 1: The abstract as a whole is clearly formulated and gives an idea of the content of the article.

Point 2: The second paragraph of the introduction - lines 40-41 - it is desirable to add appropriate references, for example, "for example by plasma etching [***], laser micro abrading [***], ..."

Response 2: The second paragraph, indeed, do not have any annotations to relevant references, because it is in some way preface to next paragraphs, which comprises adequate reference annotations.

Point 3: Line 44 and then - I recommend that you specify a reference immediately after the author of "Drummer et al. [***] used the ..."…” Rodríguez-Vidal et al. [***]…” etc.

Response 3: The changes have been made in the manuscript and is marked in yellow. Lines no. 45; 66; 199.

Point 4: Line 115 - not sure if quotes are needed in the word “filled”

Response 4: The change has been made in the manuscript and is marked in yellow. Line no. 116

Rivet position was counted from the edge of the metal insert that was first filled by a molten polymer.

Point 5: Figure 1.а - technical image quality should be improved.

Response 5: The correction in the figure has been done and introduced into the manuscript.

Point 6: Figure 3. - at least image a is not in my opinion redundant. In principle, the general view of the equipment does not give much to the reader. A circuit diagram would be preferable.

Response 6: As in general viewpoint, we agree with the Reviewer. However, we have decided to insert the pictures of the IR equipment and mounted specimen, primarily to gain the expression of necessity of sample preparation with black colorants, what enabled thermography observation.

Point 7: Table 1. - it is desirable to use a decimal point, not a comma to indicate the decimal point.

Response 7: The changes have been made in the manuscript and is marked in yellow, in Table 1.

Point 8: Figure 4. - I recommend increasing the magnification - now these images do not provide much information.

Response 8: Figure 4 was replaced in revised version by new one.

Point 9: A large number of quoted terms is not very good for a scientific style ... “rivet local”, so-called “classic” fountain (it is not clear why “so-called” - who calls it that), so-called “touch-off” , “Frozen”, in such “bi material coupling” - in most cases, quotation marks are simply superfluous.

Response 9: All quotation marks mentioned in Point 9 have been removed. The changes have been made in the manuscript and is marked in yellow

Point 10: Figure 5 - image quality should be improved

Response 10: The correction in the figure have been done and introduced into the manuscript. In revised version Figure 5 is presented as a Figure 4.

Reviewer 3 Report

The idea behind the study is interesting but the manuscript has some important shortcomings which, to the opinion of the reviewer, make it unpublishable in the present form:

  • Lack of results: The paper contains an insufficient amount of results to be considered as a full-length research paper. Basically, only two quantitative results are presented: the maximum force needed to break the samples (Table 1) and the temperature evolution before braking of the samples (Figure 6). Images of worn samples in Figures 4 and 5 are of poor quality and do not provide any relevant information. Also, the discussion of results is too general and not supported with sufficient argumentation or references to other relevant studies. The authors need to extend the Results section correspondingly and provide additional and more detailed results and analyses on the subject of matter. Without this, to the opinion of the reviewer, the paper cannot be published as a full-length research paper.
  • Grammar and writing: There are quite a lot of grammar and writing mistakes and sometimes sentences are difficult to understand. A thorough revision of writing is suggested. The help of third reader – possibly a native English speaker or a professional proofreading service – is strongly recommended; however, if no such possibility exists, the use of writing assistant programs such as e.g. Grammarly is advised.

It is suggested the authors address the issues raised by the reviewer and modify the manuscript accordingly.

For assistance, some common grammar mistakes and other ambiguities are outlined below:

  • Often singular and plural forms are not used correctly, e.g. line 14 “the metals inserts was joined”, line 23 “3 rivets was used”, line 119 “mechanical data’s will be considered”, etc.
  • Often some sentences or parts of sentences are difficult to understand, e.g. line 20 “a cut-off the rivets mechanism determining the sample damage”, line 165 “realized by tensile”, lines 298-299 “is significantly the rivets number dependent”, etc.
  • Abbreviations should be introduced the first time, the abbreviated term is mentioned. For polyamide 6, this should be already after line 14. Also, once abbreviation is introduced, it should be used consistently throughout the text. Currently, “polyamide 6” is often used in full-word throughout the text (lines 72, 123, 294, etc.)
  • The symbol used for degrees Celsius does not seem to be correct – its position and size are unusual.
  • Line 63: Instead of “pattering” it should be “patterning”.
  • Lines 91-92: If using “either-or” statements, the “or” part should be added as well.
  • Line 125: Please describe what the specific technical terms such as talc, rivets, etc. represent as this may not be clear to all the readers.
  • Line 129. The term “respectively” should be used after listing the respective units (not before).
  • Uniform style should be used for “polymer-metal”, currently different styles are used (see lines 145, 188, etc.)
  • Point should be used as decimal separator (see e.g. Table 1 and lines 168 and 265).
  • Line 221: The term “diminishing” should be replaced with e.g. “decreasing in size”.
  • Lines 226, 242: The term “fountain flow” is unusual; did the authors wish to refer to “laminar flow”?
  • Line 227: Instead of “happens” the term “occurs” should be used.
  • Line 228: Why reference [27] is positioned in the middle of the sentence and what does it refer to?
  • In Figure 4, “rivett” is written with double t, while in the text it is written with single t.
  • Line 247: Instead of “talk” it should be “talc”.
  • Instead of “on Figure”, it should be “in Figure” – see e.g. lines 239 and 257. This should be corrected throughout the entire manuscript.
  • Instead of “this”, it should be “thus” – see e.g. lines 222 and 261. This should be corrected throughout the entire manuscript.
  • Line 275: It is unusual to use abbreviations such as “comp.” in a scientific paper.
  • In the conclusions, the information that the “cross section lower than 30% of specimen pristine cross section is the most valuable” is presented for the first time.
  • Lines 281-283 and 303-306: Using thermovision for the detection of premature effects of polymer-metal damage is probably useless since the temperature increase would occur only if the speed of the deformation would be sufficiently high. At the same time, the temperature increase would be observed only shortly before braking of the sample, therefore, not providing any possibility for reaction or repair before the failure of the component.

Author Response

Response to Reviewer 3 Comments

The idea behind the study is interesting but the manuscript has some important shortcomings which, to the opinion of the reviewer, make it unpublishable in the present form:

Point 1: Lack of results: The paper contains an insufficient amount of results to be considered as a full-length research paper. Basically, only two quantitative results are presented: the maximum force needed to break the samples (Table 1) and the temperature evolution before braking of the samples (Figure 6). Images of worn samples in Figures 4 and 5 are of poor quality and do not provide any relevant information. Also, the discussion of results is too general and not supported with sufficient argumentation or references to other relevant studies. The authors need to extend the Results section correspondingly and provide additional and more detailed results and analyses on the subject of matter. Without this, to the opinion of the reviewer, the paper cannot be published as a full-length research paper.

Response 1: Some of presented results due to their less informative character were removed (Figure 4 in past version), and replaced by the improved versions (Figure 4 in present version). Last figure (Figure 6) in revised version of the manuscript was added, due to its relation to the thermographic investigation techniques. The authors decided to extend the information concerning the IR thermography investigation methodology because the detailed literature survey, where mechanical method, modeling (CATIA, ARAMIS) analysis, are often explored, shown that those methods are used rare in revealing the deformation nature of riveted hybrid elements.

Discussion of results was also extended by the new argumentation connected with cited references, and it was marked in yellow.

Point 2: Grammar and writing: There are quite a lot of grammar and writing mistakes and sometimes sentences are difficult to understand. A thorough revision of writing is suggested. The help of third reader – possibly a native English speaker or a professional proofreading service – is strongly recommended; however, if no such possibility exists, the use of writing assistant programs such as e.g. Grammarly is advised.

Response 2: Grammar check of the whole manuscript has been performed by MDPI English Editing service.   

It is suggested the authors address the issues raised by the reviewer and modify the manuscript accordingly.

For assistance, some common grammar mistakes and other ambiguities are outlined below:

Point 3: Often singular and plural forms are not used correctly, e.g. line 14 “the metals inserts was joined”, line 23 “3 rivets was used”, line 119 “mechanical data’s will be considered”, etc.

Response 3: The grammar corrections have been made in the manuscript and are marked in yellow

Line no. 14 In this, work, metal inserts were joined with polyamide 6 by using the injection-molding technique.

Line no. 25 The maximal values of the joint’s failure force under tensile-shear tests were achieved for samples where three rivets were used.

Line no. 121 The mechanical data were considered in relation to the corresponding IR thermograms.

Point 4: Often some sentences or parts of sentences are difficult to understand, e.g. line 20 “a cut-off the rivets mechanism determining the sample damage”, line 165 “realized by tensile”, lines 298-299 “is significantly the rivets number dependent”, etc.

Response 4: The corrections have been made in the manuscript and are marked in yellow

Lines no. 21,22 The study showed that, for polymer–metal joined samples with fewer than three rivets, the destruction of rivets by cutting meant sample damage. 

Line no. 167 Samples with single lap-joint geometry were designated to determine the strength of joints by shear test at a crosshead speed of 2 mm/min.

Lines no. 318-320 The most pronounced observation coming out from our investigations is that the nature of fracture mechanics, of injection molded plastic–metal parts joined with rivets, is considerably dependent on rivet number.

Point 5: Abbreviations should be introduced the first time, the abbreviated term is mentioned. For polyamide 6, this should be already after line 14. Also, once abbreviation is introduced, it should be used consistently throughout the text. Currently, “polyamide 6” is often used in full-word throughout the text (lines 72, 123, 294, etc.)

Response 5: The corrections have been made in the manuscript and are marked in yellow

Line no. 16 The metal parts, made of steel grade DC 04, were mechanically interlocked with polyamide 6 (PA6) by rivets as mechanical connection between both partners in the form of polymer filling the holes in the metallic parts.

Line no. 74 Another approach for microstructuring the hot-rolled AlMg3 alloy by electrochemical treatment, before joining with PA6 filled with 30 wt % of glass fiber, was presented in [20].

Line no. 124 The PA6 injection molding grade Tarnamid T-27, with a melt volume flow index (MFI) of 120 cm3/10 min at 275 OC (supplied by Grupa Azoty, Poland) and its composite with talc, a mineral filler with a shallow form, suitable to use as markers of the liquid-flow lines, were applied in our investigations.

Line no. 316 In this work, we tested mechanically coupled pieces/elements made of two different materials, PA6 and steel.

Point 6: The symbol used for degrees Celsius does not seem to be correct – its position and size are unusual.

Response 6: The corrections of the degree Celsius symbol have been made in the manuscript and are marked in yellow

Point 7: Line 63: Instead of “pattering” it should be “patterning”.

Response 7: The correction has been made in the manuscript and is marked in yellow, line no.64

Some papers were also devoted to presenting the laser-micropatterning technique, used as a novel tool to enhance micromechanical adhesion between metal and polymer, subjected to overmolding by injection [16,17].

Point 8: Lines 91-92: If using “either-or” statements, the “or” part should be added as well.

Response 8: The correction has been made in the manuscript, new sentences in revised version overtyped the sentences in lines 94-95.

Point 9: Line 125: Please describe what the specific technical terms such as talc, rivets, etc. represent as this may not be clear to all the readers.

Response 9: The corrections have been made in the manuscript and is marked in yellow,

Lines no. 15-17,

The metal parts, made of steel grade DC 04, were mechanically interlocked with polyamide 6 (PA6) by rivets as mechanical connection between both partners in the form of polymer filling the holes in the metallic parts.

Lines range 124-127

The PA6 injection molding grade Tarnamid T-27, with a melt volume flow index (MFI) of 120 cm3/10 min at 275 OC (supplied by Grupa Azoty, Poland) and its composite with talc, a mineral filler with a shallow form, suitable to use as markers of the liquid-flow lines, were applied in our investigations.

Point 10: Line 129. The term “respectively” should be used after listing the respective units (not before).

Response 10: The correction has been made in the manuscript and is marked in yellow

Line no. 131 The dimensions of the single metal insert were 75 x 10 mm (length x width, respectively), and thickness was 1 mm..

Point 11: Uniform style should be used for “polymer-metal”, currently different styles are used (see lines 145, 188, etc.)

Response 11: The corrections have been made in the manuscript and are marked in yellow, lines no. 147, 187

Point 12: Point should be used as decimal separator (see e.g. Table 1 and lines 168 and 265).

Response 12: The corrections have been made in the manuscript and are marked in yellow,

Table 1 and lines no. 144, 169, 258

Point 13: Line 221: The term “diminishing” should be replaced with e.g. “decreasing in size”.

Response 13: The correction has been made in the manuscript and is marked in yellow, line no. 214

This effect may have led to the fracture at the bottom edge of the rivet, whereas the cross-section continuously decreased in size due to the significant plastic deformation..

Point 14: Lines 226, 242: The term “fountain flow” is unusual; did the authors wish to refer to “laminar flow”?

Response 14: In our opinion, fountain flow is preferred term, because  of the nature of polymer flow with the fountain – like velocity vectors distribution on the channel cross-section, where the flow vector at the center is divided into flow lines perpendicular, followed by parallel positions in the near wall layer, as it is often presented in injection molding flow analysis, as it was newly ascribed by Jong W.R. et al. (Figure below).

Figure of polymer flow behavior presented by Jong W.R.; Hwang S.S.; Wu C.C.; Kao C.H.; Huang Y.M.; Tsai M.C., Using a Visualization Mold to Discuss the Influence of Gas Counter Pressure and Mold Temperature on The Fountain Flow Effect, Inter. Polym. Proc., 2018, 33, 255-267,

DOI 10.3139/217.3496  

Point 15: Line 227: Instead of “happens” the term “occurs” should be used.

Response 15: The correction has been made in the manuscript and is marked in yellow, line no. 220

This effect may be related to the disarrangement of fountain flow in the proximity of rivets [28,29], i.e., the local disorientation of the macromolecular structure by molten-polymer filling the cylindrical holes may occur.

Point 16: Line 228: Why reference [27] is positioned in the middle of the sentence and what does it refer to?

Response 16: The reference [27] was deleted from this sentence.

Point 17: In Figure 4, “rivett” is written with double t, while in the text it is written with single t.

Response 17: Figure 4 was replaced by new one.

Point 18: Line 247: Instead of “talk” it should be “talc”.

Response 18: The correction has been made in the manuscript and is marked in yellow, line no. 231

Point 19: Instead of “on Figure”, it should be “in Figure” – see e.g. lines 239 and 257. This should be corrected throughout the entire manuscript.

Response 19: The corrections have been made in the manuscript and are marked in yellow, lines no. 150, 237, 251

Point 20: Instead of “this”, it should be “thus” – see e.g. lines 222 and 261. This should be corrected throughout the entire manuscript.

Response 20: The corrections have been made in the manuscript and are marked in yellow, lines no. 215, 255

Point 21: Line 275: It is unusual to use abbreviations such as “comp.” in a scientific paper.

Response 21: Mentioned  abbreviation was removed from this sentence.

Point 22: In the conclusions, the information that the “cross section lower than 30% of specimen pristine cross section is the most valuable” is presented for the first time.

Response 22: The information was removed from this sentence.

Point 23: Lines 281-283 and 303-306: Using thermovision for the detection of premature effects of polymer-metal damage is probably useless since the temperature increase would occur only if the speed of the deformation would be sufficiently high. At the same time, the temperature increase would be observed only shortly before braking of the sample, therefore, not providing any possibility for reaction or repair before the failure of the component.

Response 23:  We assumed, that not only temperature rise could be helpful in detection of polymer-metal damage, but also change in the signal nature. Additional comment was paste in the text, and is marked in yellow, lines range 295-299, 3236-324.

Round 2

Reviewer 1 Report

The paper has been well revised. Only a few minor corrections are needed:

Line 130: En 10130 or EN 10130?

Line 133: The comma is missing: 1, 2 3, or 4,

Tab. 1 Stand. Dev. (MPa) - different font size? please check

The text contains references to Figure 1 and Figure 3, but the partial figures Figure 1a (3a) and Figure 1b (3b) are not mentioned separately in the text. The same applies to figure 4a. In contrast, all parts of figures 5 and 6 are referred in the text.

Author Response

Response to Reviewer 1 Comments after 2nd Revision

The paper has been well revised. Only a few minor corrections are needed:

Point 1: Line 130: En 10130 or EN 10130?.

Response 1: The change has been made in the manuscript. Line no. 130

Point 2: Line 133: The comma is missing: 1, 2 3, or 4,

Response 2: The change has been made in the manuscript. Line no. 133

Point 3: Tab. 1 Stand. Dev. (MPa) - different font size? please check

Response 3: The font size has been corrected in Table no 1.

Point 4: The text contains references to Figure 1 and Figure 3, but the partial figures Figure 1a (3a) and Figure 1b (3b) are not mentioned separately in the text. The same applies to figure 4a. In contrast, all parts of figures 5 and 6 are referred in the text.

Response 4: The references to all partial figures have been mentioned separately in the text. Lines no. 135, 169, 173, 215, 217.

Reviewer 3 Report

The authors have addressed the issues raised by the reviewer, extended the manuscript accordingly and applied the necessary corrections. The paper is now significantly improved. However, in the conclusions, some statements which are not supported by the actual study are still provided. It is suggested that the authors revise these accordingly:

  • Lines 327-328: The term “proven tool” is misleading, because, in the present study, IR thermography was not used in on-field tests to reveal the premature effects of polymer-metal damage processes, but under controlled laboratory conditions under specific testing conditions which may not necessarily correspond to the real-life structures. Therefore, IR thermography is not yet a “proven tool” for use in the suggested applications. Consequently, the term “proven” should be omitted. The same is true for the term “certainly convenient” on line 326 – it is premature to make this type of statements based on a few laboratory tests. Please, mitigate the statements accordingly.

At the same time, in the revised manuscript minor writing and grammar mistakes could be found as well. It is suggested that the authors apply these to further improve the quality of the manuscript:

  • Line 39: Instead of “were done”, it should be “were performed”
  • Lines 68-70: It should be “…the impact of pulses with a length of nanoseconds was investigated, as well as the impact of…”
  • Line 106: It should be “from one material to another”
  • Line 114: Instead of “to register”, it should be “to record”
  • Line 133: There is a comma missing between numbers 2 and 3
  • Line 139: Instead of “polymer/metal” it should be ““polymer–metal”
  • Line 234: Instead of “talclike”, it should be “talc-like”
  • Line 234: Instead of “10 wt”, it should be “10 wt%”
  • Line 238-239: Instead of “…placed perpendicularly talc particles, and almost parallel to the flow direction…”, it should be “…talc particles, placed perpendicularly and almost parallel to the flow direction…”
  • Line 239: Instead of “fountainlike”, it should be “fountain-like”
  • Lie 317: Instead of “rivetlike”, it should be “rivet-like”
  • Line 321: Instead of “dependent on rivet number”, it should be “dependent on the number of rivets”
  • Line 322: Instead of “is the most valuable”, it should be “are the most valuable”
  • Line 324: Instead of “enable to”, it should be “enables to”

Author Response

Response to Reviewer 3 Comments after 2nd Revision

Dear Reviewer,

Thank you for Your insightful and precise review of our work, which surely contributed to a better understanding of the scientific problems related to the subject of the publication and will help with the elimination of potential errors in the future.

We would also like to express our gratitude for the revision of our manuscript and the opportunity to re-submit it, incorporating all of mentioned suggestions.

Our comments are noted below.

Point 1: Lines 327-328: The term “proven tool” is misleading, because, in the present study, IR thermography was not used in on-field tests to reveal the premature effects of polymer-metal damage processes, but under controlled laboratory conditions under specific testing conditions which may not necessarily correspond to the real-life structures. Therefore, IR thermography is not yet a “proven tool” for use in the suggested applications. Consequently, the term “proven” should be omitted. The same is true for the term “certainly convenient” on line 326 – it is premature to make this type of statements based on a few laboratory tests. Please, mitigate the statements accordingly.

Response 1: The appropriate changes have been proposed:

proven tool -  supporting tool,

certainly convenient -   helpful,

and they are paste in the manuscript. Lines no. 329 and 331.

At the same time, in the revised manuscript minor writing and grammar mistakes could be found as well. It is suggested that the authors apply these to further improve the quality of the manuscript:

Point 2: Line 39: Instead of “were done”, it should be “were performed”

Response 2: The change has been made in the manuscript. Line no. 39.

Point 3: Lines 68-70: It should be “…the impact of pulses with a length of nanoseconds was investigated, as well as the impact of…”

Response 3: The changes have been made in the manuscript. Lines no. 68 to 70.

Point 4: Line 106: It should be “from one material to another”

Response 4: The change has been made in the manuscript. Line no. 108.

Point 5: Line 114: Instead of “to register”, it should be “to record”

Response 5: The change has been made in the manuscript. Line no. 115.

Point 6: Line 133: There is a comma missing between numbers 2 and 3

Response 6: The comma was inserted in the text. Line no. 135.

Point 7: Line 139: Instead of “polymer/metal” it should be ““polymer–metal”

Response 7: The correction has been made in the manuscript. Line no. 141.

Point 8: Line 234: Instead of “talclike”, it should be “talc-like”

Response 8: The correction has been made in the manuscript. Line no. 237.

Point 9: Line 234: Instead of “10 wt”, it should be “10 wt%”

Response 9: The correction has been made in the manuscript. Line no. 238.

Point 10: Line 238-239: Instead of “…placed perpendicularly talc particles, and almost parallel to the flow direction…”, it should be “…talc particles, placed perpendicularly and almost parallel to the flow direction…”

Response 10: The change has been made in the manuscript. Line no. 241.

Point 11: Line 239: Instead of “fountainlike”, it should be “fountain-like”

Response 11: The correction has been made in the manuscript. Line no. 243.

Point 12: Lie 317: Instead of “rivetlike”, it should be “rivet-like”

Response 12: The correction has been made in the manuscript. Line no. 320.

Point 13: Line 321: Instead of “dependent on rivet number”, it should be “dependent on the number of rivets”

Response 13: The change has been made in the manuscript. Line no. 324.

Point 14: Line 322: Instead of “is the most valuable”, it should be “are the most valuable”

Response 14: The change has been made in the manuscript. Line no. 325.

Point 15: Line 324: Instead of “enable to”, it should be “enables to”

Response 15: The change has been made in the manuscript. Line no. 327.